

**The hidden ecological resource of andic soils in mountain ecosystems:**
**evidences from Italy**
Fabio Terribile[1,2], Michela Iamarino[1*], Giuliano Langella[3], Piero Manna[3],
Florindo Antonio Mileti[1], Simona Vingiani[1,2], Angelo Basile[2,3]
[1]*Department of Agricultural Sciences, University of Naples Federico II, Via Università*
*100, 80055 Portici (Napoli), Italy*
[2] *Interdepartmental Research Centre CRISP, University of Naples Federico II, Via*
*Università 100, 80055 Portici (Napoli), Italy*
[3] *CNR ISAFOM, Via Patacca 85, 80056 Ercolano (Napoli), Italy*
*corresponding author: terribilesci@gmail.com
**Abstract**
Andic soils have unique morphological, physical and chemical properties that induce
both considerable soil fertility and great vulnerability to land degradation. Moreover
they are the most striking mineral soils in terms of large organic C storage and long C
residence time; this is especially related to the presence of poorly crystalline clay
minerals and metal-humus complexes. Recognition of these soils is then very important.
Here we attempt to show, through the combined analysis of 35 sampling points chosen,
throughout the Italian non volcanic mountain landscapes, in accordance to specific
physical and vegetation rules, that soils rich in poorly crystalline clay minerals have an
utmost ecological importance.



More specifically, in various non-volcanic mountain ecosystems (>700 m) and in low
slope gradient locations (<12°), in agreement to recent findings, we found the
widespread occurrence of soils with andic features having distinctive physical and
hydrological properties including low bulk density and remarkable high water retention.
Furthermore, we show a demonstration of the ability of these soils to affect ecosystem
functions by analysing their influence on the timescale acceleration of photosynthesis
estimated by NDVI measurements.
Our results are hoped to be a starting point for better understanding the ecological
importance of andic soils and also possibly to better consider pedological information in
C balance calculations.
**Keywords: fertile soils, high carbon storage capability, NDVI measurements,**
**hydrological properties, Andosols**



**1.    Introduction**
Soils having andic features (allophanic and non-allophanic) are known to have a unique
set of soil morphological, physical and chemical properties. Between them (i) high
porosity (bulk density generally $< 0.90$ g cm$^{-3}$), (ii) friable structure, (iii) high water
retention capacity, (iv) large reserves of easily weatherable minerals, (v) high
susceptibility to liquefaction, etc. Moreover, between all mineral soils, those with andic
features have the largest C storage capacity and long C residence time (Post, 1983;
Batjes, 1996; Amundson, 2001), which can be ascribed to the presence of poorly
crystalline clay minerals (Basile-Doelsch et al., 2005) and fungal and arthropodal SOM
(Nierop et al., 2005), but also to the specific physical and chemical properties that make
these soils some of the world's most fertile (Leamy, 1984; Shoji et al., 1993; McDaniel
et al., 2005). Despite these characteristics associated to C storage, andic features are
simply not considered in global carbon balance estimates (e.g. IPCC, 2006; Luo et al.,
2015); in fact in these estimates - in the best of cases - the contribution of soils (Parton
et al., 1987) is limited to organic C and soil texture parameters ignoring both other
important chemical and physical properties and the occurrence of well-known analytical
artefact in using texture data on soils difficult to disperse such as those having andic or
oxic features (Bartoli et al., 1991).
This lack of acknowledgment of andic soils is becoming more important considering
that in recent years soils with andic features have been found, along with well
established volcanic landscapes (Shoji et al., 1993; Arnalds & Stahr, 2004; Lulli, 2007),
in many "non-volcanic" mountain ecosystems (NVME) throughout the world (e.g.
Baumler et al., 2005; Dümig et al., 2008; Iamarino & Terribile, 2008; Scarciglia et al.,
2008; Graham & O'Geen, 2009; Rasmussen et al., 2010; McDaniel & Hipple, 2010;



Vingiani et al., 2014; Estevez et al., 2016). Given that two or three times more C is
stored in soils (Dixon et al., 1994) than occurs in the atmosphere as $CO_2$ and that andic
soils have such important C storage abilities (Torn et al., 1997), the above lack of
acknowledge of andic soils in carbon balance estimates is indeed unfortunate.
Moreover, in view of their large C storage capability, the danger of degradation of andic
soil is indeed high because they are some of the most vulnerable soils in the world in
terms of soil erosion (Arnalds, 2001) and rapid flow landslides (Basile et al. 2003;
Terribile et al. 2007; Vingiani et al., 2015).
**1.1. Aim and rationale**
All the above shows the need for a much better understanding about the importance of
andic soils and their ecological role. In this context, the aim of this contribution is to
attempt an insight about the influence of andic soil in Italian NVME over (i) vegetation,
through remotely sensed vegetation indexes and (ii) soil hydrological properties of
utmost importance for plant growth.
To achieve the above, a combined approach has been undertaken evaluating both 35
soils having different degrees of andic features in NVME (Figure 1) and the NDVI
dynamics of their sites.
All sites were chosen in order to select mountain soils (> 700 m asl) in conservative
geomorphological settings (slope gradient < 12°) and in areas with high primary
productivity (estimated using time series max NDVI value) from different parts of Italy
(see methods and Iamarino and Terribile, 2008).
The background of this approach being that (i) the above environmental factors can
promote andosolization and (ii) most importantly, that the great fertility of soils with



andic features positively affects plant primary productivity in natural ecosystems. Hence
the use of remotely sensed vegetation indexes (i.e. NDVI, EVI, etc.) can be a valuable
tool to address this topic: NDVI (Rouse at al., 1973) is strongly related to
photosynthetic activity and has been widely used to estimate landscape patterns of
primary production (Wang at al., 2004; Fensholt et al., 2012) and even net primary
production (Tucker & Sellers, 1986). Moreover, time series of NDVI and the related
NDVI metrics have proved to be a powerful tool for addressing plant dynamics and
yield prediction in both agriculture and natural ecosystems at different scales (Reed et
al., 1994; Zhang et al., 2003; Bolton & Friedl, 2013).
**2.      Materials**
**2.1. Study site**
This specific work refers to the whole Italian mountain territory (Figure 1). Italy
develops between the 35° and 47° North parallel and it is located in the middle of the
temperate zone of the Northern Hemisphere. It has an extremely articulated territory; 2
major mountain chains occupy more than 35% of the entire national surface: (i) the
Apennines, with predominantly sedimentary rocks, crossing almost entirely the Italian
territory  from S to N, with altitude reaching 2900 m asl (Gran Sasso); (ii) the Alps,
having predominantly metamorphic and igneous rocks, separating Italy from the rest of
Europe, with maximum altitude over 4,000 m asl (Monte Bianco, Monte Rosa,
Matterhorn). The remaining territory is mainly occupied by hilly systems (about 40%)
including those portions of Apennines slowly degrading towards  the sea, both at E and
W. Plain systems only occupy just over 20% of the entire territory.



In general terms the climate - known to be mild – is heavily influenced by the sea. With
respect to Italian mountain areas it can be assumed that for soil climate (Soil Survey
Staff, 2014) the mean moisture regime is udic (it may become ustic at lower elevation)
whereas the mean temperature regime is generally mesic (it may become frigid and
cryic at high elevation) (Costantini et al., 2004; Costantini et al., 2013).
**2.2 Soil sampling**
Soil sampling was designed to collect fertile mountain soils in conservative
geomorphological settings from different parts of Italy. The soils were sampled from (i)
mountain environments (> 600 m asl estimated by a 270 m spatial resolution DEM
obtained from the Italian Geological Service), (ii) geomorphological conservative
landscapes with moderately low slopes (slope gradient < 30° evaluated by the DEM) to
minimise the risk of sampling eroded soils and finally (iii) areas with high primary
productivity estimated using the max NDVI value (NDVI threshold 0.65) obtained from
MODIS Images MVC (230 m spatial resolution) for the period 28/7 - 13/8 2014 (which
is a strong vegetative growth period in Italy). Morphological and chemical (aggregated)
data of these pedons (28 soils after the selection reported in paragraph 2.3) along with
the background to this methodology are given in Iamarino and Terribile, 2008.
These information were further supplemented with data of 7 soils: 4 newly surveyed
and analysed soils, 3 soils reported in the scientific literature and consistent with the
previously stated rules: 1 soil concern research work in the Abruzzo region (Frezzotti
and Narcisi, 1996), 1 soil the CON.ECO.FOR program (Corpo Forestale dello Stato,
2003), a further soil was retrieved from the ISRIC database (ISRIC, 2005).



**2.3 NDVI and land use data**
In-depth analysis on time-based NDVI was performed using a MODIS VI algorithm
which operates on a per-pixel basis and relies on multiple observations over a 16-day
period to generate a maximum composite VI MVC. In order to extract the NDVI
metrics (maximum NDVI, integrated NDVI sum over the growing period, acceleration
of photosynthesis or rate of green-up, NDVI derivatives) some pre-processing of the
data were necessary (i.e. cloud contamination) following established procedures (Reed
et al., 1994). After such processing, about 15% of the NDVI observations had to be
discarded and the whole dataset was excluded from this work. This is related to well-
known problems in remote sensing, due to high and persistent cloud contamination and
in some cases also to the presence of rock outcrops inside the area of the investigated
pixels.
NDVI data were chosen to incorporate years having marked contrasting climate and
then - potentially - contrasting vegetation indexes trends and metric. Analysing the
climatic database published by the Italian Ministry of Environment for the whole
country (http://www.isprambiente.gov.it/), we have chosen years 2003, 2005, 2014.
These climatic years have the following trends (values below are ranked in the order
2003, 2005, 2014 respectively):
-        similar yearly mean temperature: 13°C, 12°C, 13°C;
-        evident differences in yearly mean maximum temperature, 36°C, 35°C, 33°C;
-        most importantly, marked differences in yearly cumulated rainfall, respectively
766 mm (SD: 172mm), 870 mm (SD: 231 mm), 1143mm (SD: 540mm);
-        marked differences in Standardized Precipitation Index (McKee et al.,1993),
varying in the range 0.5-0.5; 0.5-0.0; 1.0-2.0. This index is a well-known simplified



indicator for monitoring drought and periods of anomalously wet events and it shows
evident droughts for years 2003 and 2005.
The Corine land cover (CLC level 4, 5) classification (ISPRA, 2012) was used to
produce a preliminary evaluation of the main land covers. Corine land cover classes
were locally validated for each of the sampled sites. The reported land cover classes of
chestnut, beech and broadleaf oak must be considered classes of land cover where these
species are predominant but not exclusive. The grassland class refers to both continuous
and discontinuous natural grassland.
**3.    Methods**
All statistical analysis was performed using two-tailed tests; ANOVA (Tamhane
method) was performed for multiple comparisons of means. The reported test of
significance for the latitude was performed on a "metres from the equator" basis.
At each site a soil profile was opened up, described (FAO, 1990) and sampled. Bulk
samples were collected from all the soil horizons for chemical analysis. Undisturbed
soil samples for hydrological analysis were collected from the main horizons with steel
cylinders of about 200 cm$^3$.
Bulk samples after air drying were sieved to less than 2 mm and analysed (USDA-
NRCS, 2004): organic matter was determined by the Walkley & Black method;
Al/Fe/Si in the amorphous oxides/hydroxides and in the organic matter were extracted
respectively with ammonium oxalate (Feo, Alo, Sio,) treatment at pH = 3 (Schwertmann
method) and their content levels were determined by ICP-AES. Values of Al and Fe
extracted with ammonium oxalate were used to calculate the andic feature index
$Al_o + 0.5Fe_o$. Phosphate retention was determined according to the method of Blakemore.



In order to simplify the comparison between soils features and land use or NDVI
metrics it was necessary to aggregate chemical data obtaining a single representative
value for the whole soil; then the contents of $Al_o+0.5$ $Fe_o$, P retention and organic
carbon were weighted according to horizon thickness for each of the pedons. Soils were
classified using the World Reference Base (IUSS Working Group WRB, 2015).
With respect to hydrological analysis, ten experimental points of the soil water retention
curve $\theta(h)$, ranging from saturation to -30 kPa of potential, were determined through use
of the tension table and 5 points at -100, -500, -800, -1200 and -1500 kPa were
determined through use of a pressure plate apparatus (Dane and Hopmans, 2002). The
soil samples were then dismantled and dried for 24 h in the oven at 105°C in order to
determine the water content from the weight data set and the bulk density.
The water retention experimental data were parameterised according to the unimodal
$\theta(h)$ relationship proposed by van Genuchten (1980), expressed here in terms of the
scaled water content, Se, as Equation (1) below:

$$S_e = \left[1+\left(\alpha|h|\right)^n\right]^{1/n-1} \qquad (1)$$

with $S_e=(\theta-\theta_r)/(\theta_0-\theta_r)$, and in which $\alpha$ (cm$^{-1}$) and $n$ are curve shape parameters. $\theta_0$ and $\theta_r$
respectively represent the saturated water content (at $h=0$) and the residual water
content, and may either be fixed or treated as parameters to be optimized.
To obtain a synthetic description of water retention for an easy comparison with soil
chemical analysis, we used a numeric index (*IRI*) integrating the whole water retention
function (Basile et al., 2007).
The Integral Retention Index, *IRI*, is defined by:





$$IRI = \frac{1}{wp} \int_0^{wp} \theta \, d(\log_{10}|h|)$$

(2)

where $wp$=4.2 is the wilting point. This adimensional index ($0<IRI<1$) represents the
average value of the function $\theta(\log_{10}|h|)$ on the interval [0, $wp$] and allows simple
comparisons of the whole water retention by coalescing it in a single characteristic
value.
**4.    Results and Discussion**
**4.1 Soil and landscape**
The outcome results of our procedure in terms of soil analysis and WRB soil
classification (IUSS Working Group, 2015) show that Andosol and Cambisol alone
account for more than 80 % of the observations and, most interestingly, despite
differences in soil classification, in the vast majority of cases (about two-thirds) there is
a quite high content of poorly ordered clay minerals as estimated by $Al_o+0.5Fe_o$ % as
given in Figure 2 (moderate and well expressed andicity). Iamarino and Terribile (2008)
have reported further details (data reported as horizon-based means) on 42 of these
pedons proving the general absence of podsolization and depicting a scenario where
andosoliation is the main soil process.
In Table 1 are reported the main geographical and land cover features of the studied
soils along with NDVI metrics over three contrasting climatic years; the dataset shows
that Andosols, Cambisols and Phaeozems occur at similar latitudes and elevations and
beech, oak, chestnut and grassland are the main land use. More specifically, the main
land cover unit associated with Andosols and Cambisols is the beech forest but they also
occur in other land uses. Phaeozems are mostly associated with grassland.





In all years, in sites where Andosols occur the mean value of max NDVI, integrated
sum of NDVI and NDVI green-up is always the highest, as compared to other soil
classes. This finding is very interesting and it is consistent with the high fertility of
these soils. NDVI max and NDVI integrated sum (Jun-Aug) show significant
differences between the different land cover classes, following clear diversity in plant
biology.
The analysis of NDVI trend between the 3 investigated years, shows that , as expected,
NDVI max and NDVI sum values in the wetter 2014 are always higher than in the drier
2003 and 2005. Differently the NDVI green-up values are typically higher in 2003-2005
as compared to 2014 and this NDVI green-up difference is even more pronounced
moving towards the most andic soils (Andosols). All the above clearly suggest that soils
with andic features – typically having higher water storage as compared to other soils –
enabled to produce an higher green-up. Here we must also add that further analysis
would be required to evaluate at each site trends in soil water storage and temperature
before the green-up phase.
In Table 2 are reported the main features of the studied soils; the soil dataset shows that
all soils are deep, have a friable granular/crumb soil structure at the surface; moreover,
organic C, andic features (always $Al_o+0.5Fe_o \geq 0.4\%$)  and P retention range from
moderate to high. Of all the soils, Andosols, have the highest (i) soil depth, (ii)
Alo+0.5Feo % (weighted mean) and (iii) P retention % (weighted mean). Phaeozems
have the highest organic C (weighted mean) content.
Though $Al_o+0.5Fe_o$ and P retention values in Andosols differ significantly, there are no
such significant differences between the various land cover classes, suggesting that
vegetation is of little importance in determining andic features.



In general terms the investigated soils can be considered rather homogeneous in their
morphological, chemical and physical properties although they occur in very diverse
geological and climatic mountain ecosystems; a preliminary cautious estimate
(Iamarino, 2005) of their distribution in Italy has shown their presence on about $7 \times 10^5$
ha.
This finding parallel similar ones in other parts of the world where mountain soils with
andic features (not necessarily Andosols) have been reported in Bhutan (Baumler et al.
2005), in Brasil (Dumig et al., 2008),   in California (Graham & O'Geen, 2010;
Rasmussen et al., 2010), Pacific North-West USA (McDaniel & Hipple, 2010), NW
Spain (Estevez et al., 2010) and also in Italy (Iamarino & Terribile, 2008; Scarciglia et
al., 2008; Vingiani et al., 2014).
**4.2 Andic features and soil hydrology**
Given the finding on the importance of andic soils (albeit not Andosols) in Italian non-
volcanic uplands, the question is raised as to whether such andic features are also
connected to those physical properties considered of key importance for plant growth,
namely bulk density and water retention due to their crucial role in water availability. In
order to address this issue, a selection of undisturbed soil samples, from horizons A and
B, of the previously investigated soils were analysed. The data (in Table 3) clearly show
the occurrence of very porous soils (low bulk density) and very high water retention
capability over the complete range of pressure head values. Surface A horizons
generally have lower bulk density and higher water retention explicit by IRI than the
subsoil B horizons, which must be ascribed to the contribution of organic carbon in



improving the soil structure (Kutilek and Nielsen, 1994) and therefore increasing water
retention and decreasing bulk density.
The positive high correlation (Figure 3) between $Al_o+0.5Fe_o$ (%) and IRI indicates that
higher andic features correspond to higher integrated water retention, hence very good
soil physical properties. This result is already established (Basile et al., 2007) but only
for soils having $Al_o+0.5Fe_o$ (%) larger than 2% while there are no positive evidence for
soils having much lower $Al_o+0.5Fe_o$ content (e.g. in the range 0.4-2.0%). All the above,
emphasises that poorly ordered clay minerals greatly affect soil physical properties even
at moderate to low concentration, which in turn could greatly affect water storage and
then water availability for plant ecosystem growth.
Such finding is important because it does not refer to soils in a unique location but
rather to a large variety of soils developed at different latitude and over different
bedrocks and land uses.
**4.3 Andic features and elevation against NDVI metrics**
To investigate this question further, bivariate correlation (Table 4) and regression
analyses (Figure 4) were performed between andic features ($Al_o+0.5Fe_o$ %) and NDVI
metrics for each of the observed land cover classes. In the vast majority of climatic
years and land cover classes, andic features have a positive correlation with NDVI
metrics but, generally, not significant for (i) NDVI max value and (ii) integrated sum of
NDVI (Table 4). By contrast, rather astonishing, andic features are always well
correlated with the rate of green-up (1[st] derivative of NDVI); this correlation is
significant for the driest years 2003 and 2005 and not for the wettest 2014. Highest
significant correlations are found when each land use is considered separately. For




instance, in 2003 the *r* Pearson between andic features and green-up is 0.82 for beech
and 0.83 for grassland while in the year 2005 is 0.86 for beech and 0.90 for grassland.
These results show that beech and grassland are the best performing to show the
ecological importance of andic features; furthermore, the data producing this high
correlation are spanned along a high range of $Al_o + 0.5Fe_o$ % values (see Figure 4). This
performance could be explained considering that i) beech and grassland are more
spatially homogeneous land uses as compared to oak broadleaves (e.g. oak land use is
more heterogeneous being a potential mixture of very different species sometime even
including grassland); (ii) beech and grassland land uses are less affected by strong land
management practices as compared to chestnut (in fact in the Italian landscape it is often
managed as coppice); (iii) moreover it is well known that beech is very susceptible to
severe water stress (Teissier et al., 1981).
All the above can could well explain the more responsive NDVI signal of beech and
grassland to water stress as compared to oak broadleaves and chestnut.
To the authors best knowledge, it is the first time that it is shown a close connection
between NDVI metrics and soil andic features. This result can have important
consequences in terms of better understanding the ecology of Italian mountain
ecosystems.
Differently in many different environment often it has been reported the positive
variation of NDVI against elevation (Zhan et al. 2012; Walsh et al., 2001; Chen et al.,
2006), thus since soils with andic features occur in mountain areas, it was important to
test whether the observed relationship between NDVI metrics and andic features hinder
a possibly even closer relationship between NDVI metrics and elevation.





To this respect, table 4 shows that the correlation between NDVI metrics and elevation
is much more confusing with much lower *r* values as compared with those between
NDVI and andic features. Overall both the low and negative *r* values between many
NDVI metrics and elevation show that altitude (and possibly its covariates, i.e.
temperature and rainfall) do not adequately explain variations in green active biomass
parameters. Moreover, *r* values between andic features and elevation show very low
values (e.g. $r = 0.16$ for all sites) and do not show any consistent trend (data not shown).
Then here we can state that for the first time it has been demonstrated the ecological
importance of soils with andic features over different land use canopies with respect to a
large part of Italian mountains; most probably this finding has to be connected to the
unique hydropedological properties of these soils. In fact, this result is especially
evident in the driest years (2003, 2005) while is less important in the wettest 2014 year
thus it is rather evident that the water storage of these soils may play a key controlling
role.
Our findings are also important to better acknowledge the occurrence and the
importance of these soils in C sequestration/storage estimates. Indeed, deep andic soils
(as reported in this study) have about twice (Batjes, 1996) the mean organic C content
of deep Regosols, Cambisols and Podzols which previous soil inventories (Mancini,
1966; EuDASM, 2007) considered as the main soil types in the investigated landscapes.
**5.    Conclusions**
Our study shows a close relationship between the degree of andic features and NDVI
metrics and especially with metrics describing acceleration of photosynthesis (green-





up). This finding demonstrates that there is yet much to be understood about the
ecological importance of soils in mountain ecosystem, at least for the Italian territory.
Moreover the acknowledge of the importance of these soils may also have important
consequences in terms of both soil protection in mountain environment (andic soil are
known to be easily erodible) and for better understanding the impact of climate change.
To this respect this study suggest that the unique water retention features of the andic
soils plays an important ecological role when comparing contrasting climatic years.
The above result are maybe even more pronounced considering that the current study
employed a rather simplified NDVI approach including data at coarse resolution
(MODIS) and no algorithm to mitigate the well-known saturation effect of NDVI
(Buschmann and Nagel,1993). Thus it is likely that in future, better focused studies,
may demonstrate even better and closer relationships between andic soils and green
biomass indicators.
Generally our results indicate the large potential in using remote sensed vegetation
index metrics to ameliorate soil spatial inventories. A question still arises as to whether
the general absence of strong significant correlation between andic features with both
"NDVI max" and "integrated NDVI sum" may be caused by the quoted NDVI
saturation effect.
Considering our results, it is also important to emphasise that the importance of andic
features in affecting ecosystem function is undoubtedly poorly expressed by soil
classification: in fact strict classification rules dealing with how/where to expect "andic
properties" (WRB: starting within 25 cm from the soil surface; Soil Taxonomy: within
60 cm) can lead to non-Andosols with very high andic features. However, andic
features, rather than soil class criteria, seem to better explain variability in NDVI



metrics and plant ecosystem dynamics and this finding must be of major concern for
ameliorating soil classification.
Although the importance of this key mineral soil in Italian mountain ecosystems is
demonstrated producing in turn large organic C storage and long C residence time,
proper implementation of these new data in terms of C balance calculation, reducing
uncertainties in carbon sequestration estimates and carbon sink national ecosystems
inventory, is indeed a major issue to be addressed.
Moreover, the given wide recognition of andic soils has important consequences both in
terms of C sequestration potentialities and C lost risks associated to this finding.
Suitable land management techniques are then required to match the exclusive
properties and problems connected to the presence of these soils.
Considering the many recent finds of "andic" soils worldwide, it is of great importance
to ascertain whether a wider occurrence of this hidden resource apply also to mountain
environments in other parts of the world.
Finally, we must emphasise that this study – focused on only 35 points over the Italian
landscape – is the methodological basis for producing statement at the national scale
where, accordingly, much more data are indeed required.



**Acknowledgements**
We would like to thank a number of people who helped us in (i) sampling at some of the
sites: A. Vacca, G. Maugeri, A. Mingo; (ii) performing the hydrological analysis: N.
Orefice and R. De Mascellis; (iii) producing some of the chemical analysis: L. Minieri.



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





**FIGURE CAPTIONS**

**Figure 1.** Location of the sampling points (black triangles).

**Figure 2.** Soil type (WRB classification) plotted against andic features estimated by $Al_o+0.5Fe_o$ % (weighted mean according to horizons thickness for each of the studied pedons).

The value of 0.4% in $Al_o+0.5 Fe_o$ is the "key out" requirement for entering in the Andosol (and/or Andisol) classes both in WRB and Soil Taxonomy classifications. The andic features estimated by $Al_o+0.5 Fe_o$ % can be considered weak in the range 0.4-1.0, moderate in the range 1.0-2.0 and well-expressed over 2.0.

**Figure 3.** Scatterplot between andic features estimated by $Al_o+0.5 Fe_o$ % and Integrated Retention Index (IRI). Coefficient of determination $R^2$ along with the number of data points (n) are reported.

**Figure 4.** Scatterplot between andic features estimated by $Al_o+0.5 Fe_o$ % (weighted mean $Al_o+0.5Fe_o$ % according to horizons thickness for each of the studied pedons) and the maximum value of the NDVI derivative. From left to right: grassland, beech, oak and chestnut. From bottom to top: year 2003, 2005 and 2014. The dashed lines show the linear regression for each land cover. Coefficient of determination $R^2$ along with the number of data points (n) are reported for each panel.

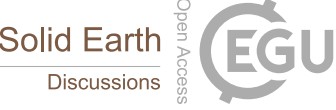


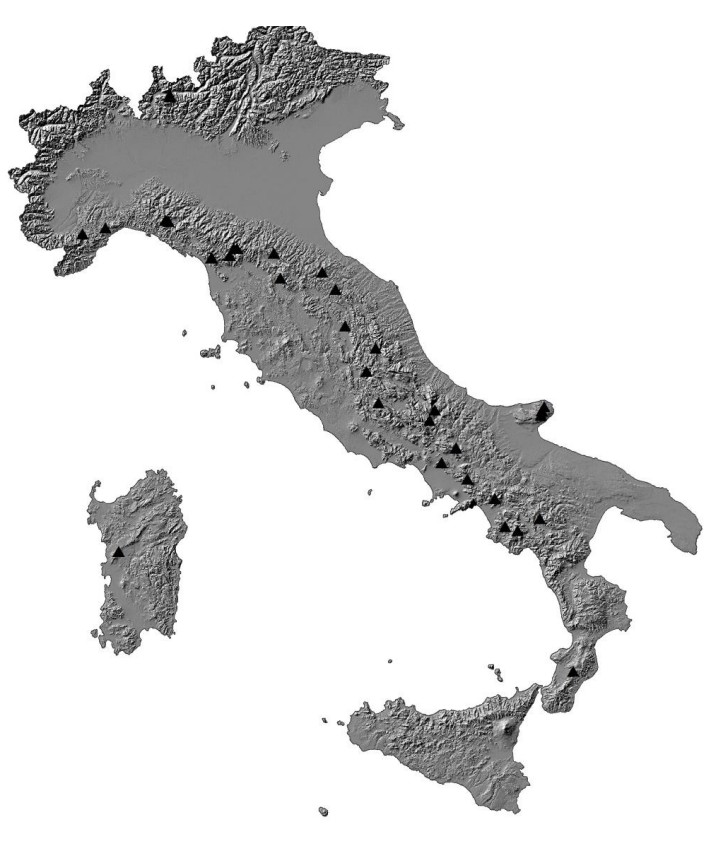

**Figure. 1**



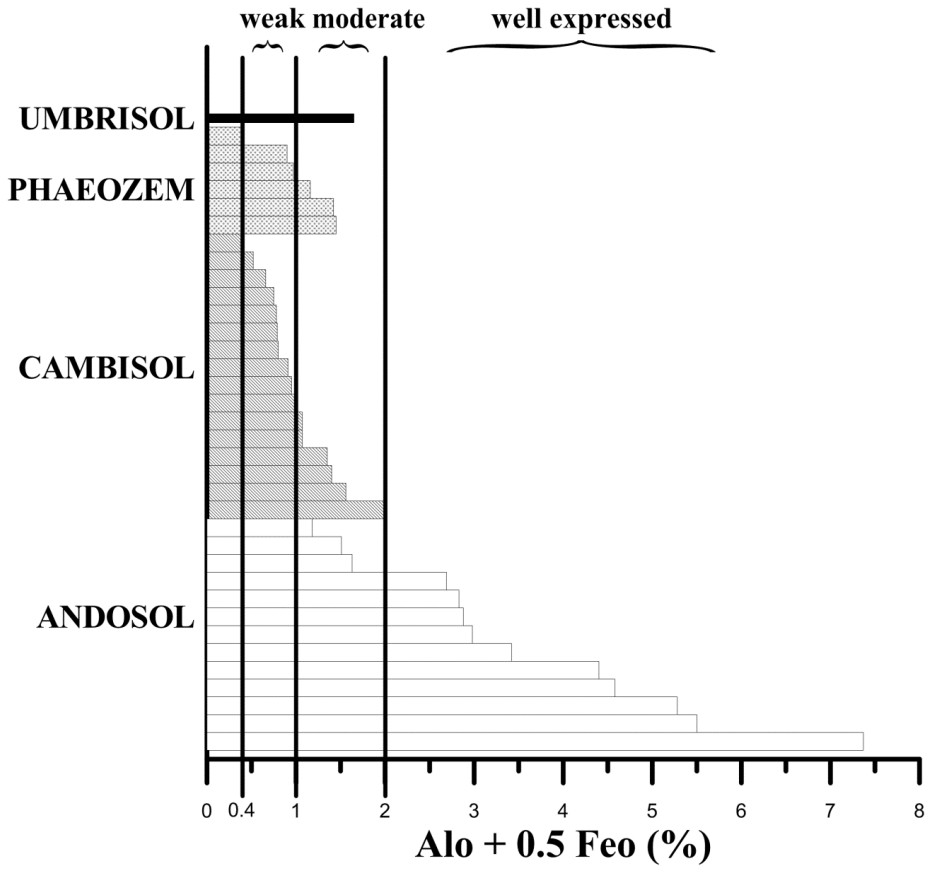

**Figure 2**





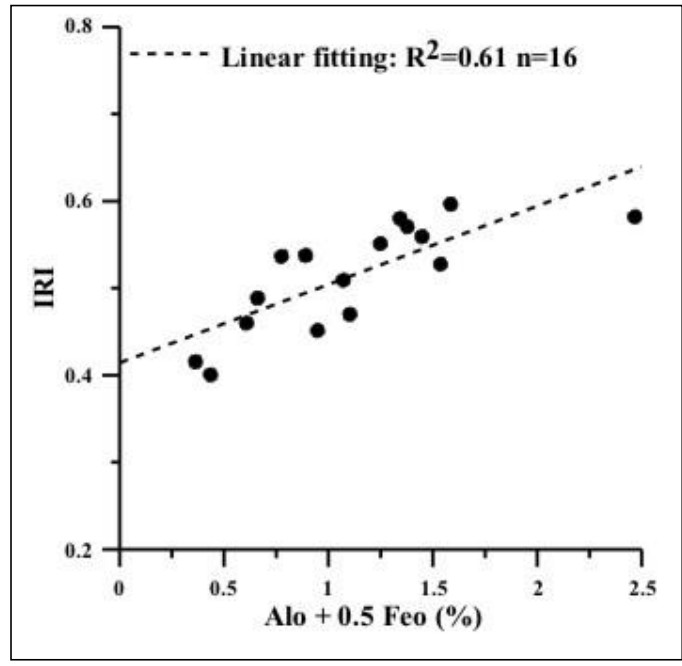

**Figure 3**



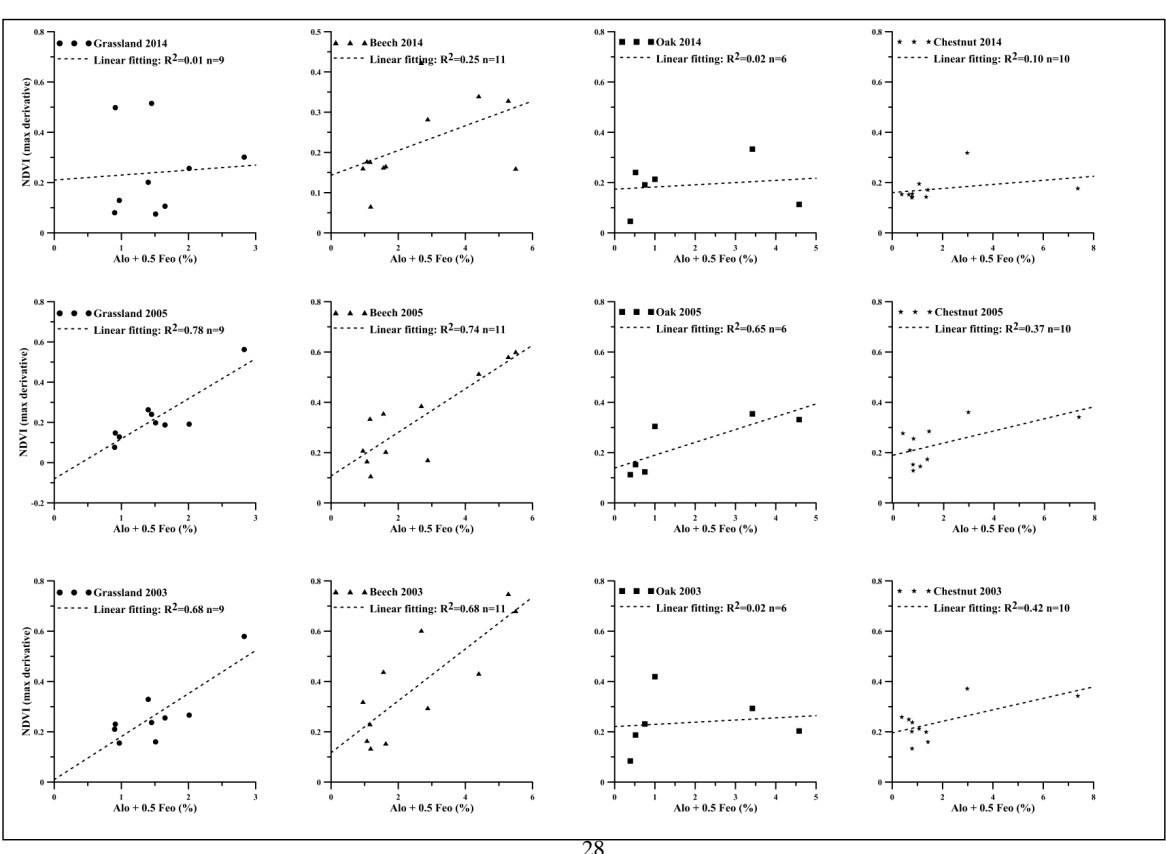

**Figure 4**


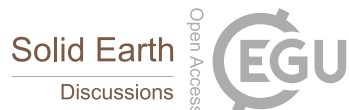

**Table 1. Main geographical and land cover features of the studied soils.** Abbrev.

| Soils/Land Cover | n. | Latitude mean | Elevation mean | Land Cover/Soils mode | NDVI max mean | | | NDVI sum (Jun-Aug) mean | | | NDVI green-up (1st derivative) mean | | |
|---|---|---|---|---|---|---|---|---|---|---|---|---|---|
| | | m | m | | 2003 | 2005 | 2014 | 2003 | 2005 | 2014 | 2003 | 2005 | 2014 |
| All soils | 35 | 4714218 N ± 187375 | 1006 ± 427 | Beech | 0.87 ±0.07 | 0.86 ± 0.09 | 0.88 ± 0.07 | 4.85 ±0.71 | 4.86 ± 0.77 | 5.06 ±0.49 | 0.29 ± 0.16 | 0.26 ± 0.14 | 0.21 ± 0.17 |
| Andosols | 13 | 4565657 N ± 144152 | 1040 ± 400 | Beech | 0.89 ±0.04 | 0.90 ±0.04 | 0.90 ± 0.05 | 5.07 ± 0.41 | 5.11 ± 0.49 | 5.24 ±0.33 | 0.38 ±0.21 | 0.36 ±0.16 | 0.24 ± 0.12 |
| Cambisols | 16 | 4828328 N ± 103489 | 943 ± 474 | Beech | 0.87 ±0.07 | 0.86 ±0.09 | 0.88 ± 0.06 | 4.97 ±0.57 | 4.97 ± 0.67 | 5.08 ±0.50 | 0.25 ±0.08 | 0.20 ±0.07 | 0.20 ± 0.09 |
| Phaeozems | 6 | 4731804 N ± 239417 | 1100 ± 427 | Grassland | 0.80 ±0.09 | 0.80 ±0.12 | 0.82 ± 0.08 | 4.02 ±1.01 | 4.03 ± 1.06 | 4.62 ±0.55 | 0.18 ±0.06 | 0.20 ±0.10 | 0.19 ± 0.17 |
| Beech | 11 | 4630565 N ± 199235 | 1219** ±291 | Cambisols Andosols | 0.92 ±0.02 | 0.92 ±0.02 | 0.92 ±0.02 | 5.28** ±0.17 | 5.36** ± 0.13 | 5.41** ±0.10 | 0.38 ±0.22 | 0.33 ±0.18 | 0.22 ±0.11 |
| Chestnut | 10 | 4829743 N ± 187599 | 680** ± 240 | Cambisols Andosols | 0.90 ±0.02 | 0.90 ±0.02 | 0.90 ±0.01 | 5.20** ±0.16 | 5.21** ± 0.11 | 5.23** ± 0.08 | 0.24 ±0.07 | 0.23 ±0.08 | 0.17 ±0.05 |
| Oak broad. | 6 | 4610094 N ± 139918 | 728 ± 424 | Cambisols Andosols Phaeozems | 0.86 ±0.07 | 0.86 ±0.06 | 0.88 ±0.05 | 4.77 ±0.71 | 4.91 ± 0.57 | 5.01 ±0.37 | 0.24 ±0.11 | 0.23 ±0.11 | 0.19 ±0.10 |
| Grassland | 8 | 4762927 N ± 111922 | 1330 ± 392 | Cambisols Andosols Phaeozems | 0.77 ±0.09 | 0.75 ±0.10 | 0.79 ±0.08 | 3.88** ±0.66 | 3.70** ± 0.74 | 4.40 ±0.56 | 0.27 ±0.14 | 0.23 ±0.15 | 0.26 ±0.17 |

** $\alpha<0.01$, * $\alpha<0.05$ (two-tailed test).

Abbr. n.: number of observations, broad.: broadleaf species.

The symbol ± after the mean value shows the standard deviation. The (n.) values refer to the number of observation available for NDVI analysis (see methods); in some sites because of strong cloud contamination not all the data could be used for NDVI analysis.

The upper part of the table refers to soil types (WRB) and the lower part refers to land cover (CORINE Land Cover classes) after site validation. NDVI MODIS metrics referring to a whole 2003, 2005,2014 time series (16 day step).



**Table 2. Main soil features of the studied soils**

| Soils / Land Cover | n. | Soil depth (solum) mean | Structure of surface A horizon mode | | Organic C mean | Alo+0.5Feo mean | P Retention mean |
|---|---|---|---|---|---|---|---|
| | | cm | | %. | ‰ | % | % |
| All soils | 35 | 88 ± 37 | Friable Gr. Cr. medium | 37 | 38.0 ± 23.0 | 2.0 ± 1.7 | 62.9 ± 26.0 |
| Andosols | 13 | 115** ± 34 | Friable Gr. Cr. medium | 69 | 45.3 ± 26.6 | 3.6** ± 1.8 | 90.2** ± 14.6 |
| Cambisols | 16 | 75 ± 31 | Friable Gr. Cr. fine; Cr. coarse | 21 | 27.3 ± 15.1 | 1.0 ± 0.4 | 46.9 ± 17.2 |
| Phaeozems | 6 | 66** ± 21 | Friable Gr. Cr. medium | 57 | 50.9 ± 22.8 | 1.0 ± 0.4 | 49.1 ± 16.5 |
| Beech | 11 | 102 ± 28 | Friable Gr. Cr. medium | 41 | 40.6 ± 22.9 | 2.6 ± 1.7 | 83.6 ± 21.6 |
| Castanea | 10 | 95 ± 38 | Friable Gr. Cr. Coarse | 40 | 23.5 ± 10.8 | 1.8 ± 2.1 | 42.7 ± 21.9 |
| Oak broad. | 6 | 70 ± 55 | Friable Gr. Cr. Fine | 25 | 34.2 ± 22.3 | 1.8 ± 1.8 | 61.3 ± 27.5 |
| Grassland | 8 | 75 ± 25 | Friable Gr. Cr. medium | 50 | 55.6 ± 25.4 | 1.5 ± 0.7 | 57.5 ± 16.7 |

** $\alpha<0.01$, * $\alpha<0.05$ (two-tailed test).
Abbr. n.: number of observations, broad.: broadleaf species, Gr.: granular, Cr.: crumb, fine: < 2 mm, medium: 2-5 mm, coarse: 5-10 mm, very coarse: > 10 mm.
The symbol ± after the mean value shows the standard deviation.
The upper part of the table refers to soil types (WRB) and the lower part refers to Land Cover (CORINE Land Cover classes) after site validation.
Chemical analyses are integrated over soil depth (solum).





**Table 3. Main physical parameters of selected soil horizons**

| Horizons | n. | Mean Bulk Density g cm$^{-3}$ | n. | Mean WC at pF=4.2 cm$^3$ cm$^{-3}$ | n. | Mean WC at pF=0 cm$^3$ cm$^{-3}$ | n. | IRI - |
|---|---|---|---|---|---|---|---|---|
| All | 35 | 0.87 ± 0.21 | 83 | 0.25 ± 0.09 | 16 | 0.79 ± 0.10 | 16 | 0.51 ± 0.06 |
| A | 16 | 0.79 ± 0.17 | 55 | 0.27 ± 0.09 | 7 | 0.85 ± 0.07 | 7 | 0.55 ± 0.04 |
| B | 19 | 0.93 ± 0.22 | 27 | 0.19 ± 0.07 | 10 | 0.75 ± 0.10 | 10 | 0.48 ± 0.06 |

Abbr. n.: number of observations, WC: volumetric water content, IRI: integrated water retention index.
The symbol ± after the mean value shows the standard deviation.
The table reports for soil horizons A and B mean bulk density, water retention at two different values of pF (0 and 4.2) corresponding to the pressure head of -0.1 and -1500 kPa, respectively, and the integrated retention index (IRI) which coalesces the water retention curve in a single value (Basile et al., 2006).



**Table 4. Bivariate correlation analysis**

**Alo+0.5 Feo (%)**

| | 2003 | | | 2005 | | | 2014 | | |
| --- | --- | --- | --- | --- | --- | --- | --- | --- | --- |
| | Mean NDVI max | Mean NDVI sum (Jun-Aug) | Mean NDVI green-up (1st derivative) | Mean NDVI max | Mean NDVI sum (Jun-Aug) | Mean NDVI green-up (1st derivative) | Mean NDVI max | Mean NDVI sum (Jun-Aug) | Mean NDVI green-up (1st derivative) |
| All sites (n.:35) | 0.19 | 0.19 | 0.61** | 0.17 | 0.20 | 0.71** | 0.16 | 0.20 | 0.23 |
| Beech | 0.36 | -0.09 | 0.82** | 0.24 | 0.20 | 0.86** | 0.42 | 0.60* | 0.50 |
| Oak | 0.16 | 0.28 | 0.14 | 0.12 | 0.29 | 0.81 | 0.20 | 0.28 | 0.15 |
| Chestnut | -0.01 | -0.21 | 0.65* | -0.13 | -0.004 | 0.61 | -0.21 | -0.01 | 0.32 |
| Grassland | -0.46 | 0.26 | 0.83* | -0.01 | 0.02 | 0.90** | -0.35 | -0.31 | 0.10 |

**Elevation**

| | 2003 | | | 2005 | | | 2014 | | |
| --- | --- | --- | --- | --- | --- | --- | --- | --- | --- |
| | Mean NDVI max | Mean NDVI sum (Jun-Aug) | Mean NDVI green-up (1st derivative) | Mean NDVI max | Mean NDVI sum (Jun-Aug) | Mean NDVI green-up (1st derivative) | Mean NDVI max | Mean NDVI sum (Jun-Aug) | Mean NDVI green-up (1st derivative) |
| All sites (n.:35) | -0.26 | -0.28 | 0.51** | -0.35* | -0.36* | 0.32 | -0.30 | -0.30 | 0.47** |
| Beech | 0.48 | 0.11 | 0.63* | 0.11 | -0.23 | 0.53 | 0.27 | 0.35 | 0.71* |
| Oak | 0.17 | 0.07 | 0.55 | 0.21 | 0.07 | 0.19 | 0.09 | 0.02 | 0.37 |
| Chestnut | -0.46 | -0.33 | 0.48 | -0.40 | -0.37 | 0.20 | -0.33 | 0.20 | 0.38 |
| Grassland | -0.46 | -0.24 | 0.36 | -0.61 | -0.42 | 0.21 | -0.49 | -0.60 | 0.26 |

** $\alpha < 0.01$, * $\alpha < 0.05$ (two-tailed test).

Correlation (r Pearson) performed between andic properties (Alo+0.5 Feo %) and NDVI metrics for each of the observed Land Cover classes (CORINE Land Cover classes) after site validation. The chemical analyses are integrated over soil depth (solum).