# Peer review of "The hidden ecological resource of andic soils in mountain ecosystems: evidences from Italy"

_Solid Earth, 2017_

## Referee Comment (RC1) · Anonymous Referee #1 · 29 Aug 2017

The paper "The hidden ecological resource of andic soils in mountain ecosystems: evidences from Italy" has an interesting way to look upon Andic soils and provide new insight in the Italian non volcanic mountain environments. It can be seen as a promising step forward in this field.

The presented manuscript is within the scope of the SE and represents a good contribution to the scientific progress. Novel data and ideas with known tools and concepts are used to provide new insight in the field. The quality and presentation of the scientific results is fair.

The methods and assumption are partly valid and not always clearly outlined. A reproduction by fellow scientist might be difficult as well as the traceability of results, due to the lack of presented data.

[Figure]

The authors do not give proper credit to mentioned methods, as several references seem to be missing. A differentiation of new/original contribution to literature data could be strongly improved, as not all data is transparently provided.

Nevertheless the title clearly reflects the content of the paper. The abstract is fairly concise. The overall presentation is still poor in structure, although version 2 is a clear improvement to version 1. The language is fluent, but in certain parts neither precise nor grammatically correct. Abbreviation, symbols etc. are mostly defined correctly.

Suggesting for certain parts of the paper, which should be clarified and reworked, are provided accordingly on the following pages of this review supplement. The number of references could be improved, as stated in the reference section.

Overall, the second version of the manuscript is a great improvement to the original version 1. Nevertheless, the manuscript is still in need of further refining.

The supplementary PDF includes a detailed list of comments for the authors, including the following main suggestions:

Several sentence structures should be shorten to improve readability.

A thorough grammar correction of the entire manuscript is needed and a re-evaluation of certain wordings.

The presented manuscript lacks throughout several section on precision and transparency regarding data and methods. It is often unclear why data is spared or used for a certain purpose, or data that is used and not provided in the manuscript in full.

Therefore it is strongly advised to refine the manuscript. Adding missing explanations together with a comprehensible reasoning of certain steps to provide a more transparent and clearer picture of the study.

Furthermore, additional references for certain themes (especially within the method and discussion section) should be added.
Tables and figures should be meticulously corrected, as they do not reflect identical data. A distinguishing between new and literature data is besides a mentioning in section 2.2, not given, and has to be improved.

Please also note the supplement to this comment:
https://www.solid-earth-discuss.net/se-2017-57/se-2017-57-RC1-supplement.pdf

───────────────────────

**Supplement:**

Comments to the Author:

The paper "The hidden ecological resource of andic soils in mountain ecosystems: evidences from Italy" has an interesting way to look upon Andic soils and provide new insight in the Italian non volcanic mountain environments. It can be seen as a promising step forward in this field.

The presented manuscript is within the scope of the SE and represents a good contribution to the scientific progress. Novel data and ideas with known tools and concepts are used to provide new insight in the field. The quality and presentation of the scientific results is fair.

The methods and assumption are partly valid and not always clearly outlined. A reproduction by fellow scientist might be difficult as well as the traceability of results, due to the lack of presented data.

The authors do not give proper credit to mentioned methods, as several references seem to be missing. A differentiation of new/original contribution to literature data could be strongly improved, as not all data is transparently provided.

Nevertheless the title clearly reflects the content of the paper. The abstract is fairly concise. The overall presentation is still poor in structure, although version 2 is a clear improvement to version 1. The language is fluent, but in certain parts neither precise nor grammatically correct. Abbreviation, symbols etc. are mostly defined correctly.

Suggesting for certain parts of the paper, which should be clarified and reworked, are provided accordingly on the following pages of this review. The number of references could be improved, as stated in the reference section.

Overall, the second version of the manuscript is a great improvement to the original version 1. Nevertheless, the manuscript is still in need of further refining. Here are some main suggestions:

- Several sentence structures should be shorten to improve readability.
- A thorough grammar correction of the entire manuscript is needed and a re-evaluation of certain wordings.
- The presented manuscript lacks throughout several section on precision and transparency regarding data and methods. It is often unclear why data is spared or used for a certain purpose, or data that is used and not provided in the manuscript in full.
- Therefore it is strongly advised to refine the manuscript. Adding missing explanations together with a comprehensible reasoning of certain steps to provide a more transparent and clearer picture of the study.
- Furthermore, additional references for certain themes (especially within the method and discussion section) should be added.
- Tables and figures should be meticulously corrected, as they do not reflect identical data. A distinguishing between new and literature data is besides a mentioning in section 2.2, not given, and has to be improved.
* * *
Detailed suggestions of specific comments and technical corrections (typing errors etc.) are market with **"page# / line#"** in respect to manuscript **version 2.**

1 / 17: Suggestion, dot after "time". New sentence starting: This feature/unique capability is especially…

1 / 18: Specify "these". e.g. delete "these" and write "andic"

1 / 19-22: sentence with 41 words should be cut in half, to improve the readability.

2 / 2: What is really define as low slope in this study? 12° or 30°, as later on page 6 line 12 written? Please clarify the used parameters

3 / 2: Please do a thorough grammar correction of the manuscript for quality reasons e.g. "an unique" not "a unique"

3 / 3: As state before, please do a sincere grammar correction e.g. delete "soil" in front of morphological and put it before properties

3/3 : Please be precise with the wording e.g. instead of "Between" use "Among or The main characteristics are.."

3 / 6-12: To improve readability, please split this sentence.

3 / 17: a single "artefact" or multiple "artefacts" ?. Also a new sentence could be started here. As a sentence with >50 words seem a bit long (line 12-18) for most readers.

3/ 22: it would be helpful for readers to write the countries and regions where the according authors (as done at 12/6-10) have found volcanic landscapes in non-volcanic mountain ecosystems, as „throughout the world" is very general. This would also emphasis the impact/importance of this research. Please state why Italy was chosen as study object, and not any of those other "around the world"?

4/1-3: Please provide clearer information about the C storage capacity. It is stated that andic soils have important C storage abilities. Which ones? How much more or longer can they store it? Why kind of soils so special?

4/20: Please clarify parameter for "low slope", 12° or 30° as later on page 6 line 12 written?

5/3: instead of ":" a dot would be sufficient

5/13: In the introduction it is explained that NVME are found around the world. Why is the focus than on Italy? Maybe start with the necessary features and narrow it than down to the best fitting location, which seems to be Italy in this case. Please provide references of the provided information within the section "study site".

5/13: Very well that there is now a map added, compared to version 1.

6/8: how was the sampling carried out. What material was used to do the sampling? An additional table including all sampling sites, with their slopes, soil type, horizon depth, elevation etc. would improve the transparency that each chosen location fulfilled the designed requirements. This is not the case yet, as only average values are presented.

6/10: Please clarify parameters. >600m or >700m as stated previously (page 2, line 1)

6/12: As mentioned early, please clarify e.g. 30° or 12°

6/18: please explain briefly the methodology. As the work in this study is with 28 samples mainly based on data of the paper "Iamarino and Terribile, 2008". It is suggested that key data, which were used for this evaluation should be presented in a referenced table.

6/21: Please do a grammar check, preposition is missing after "...and 5 soils…"

page 5-6: The clearer separation of the soil sampling and the computational part is an improvement to version 1.

7/13-24: A good, logic and clear-presented paragraph, that emphasis in detail „why", „what" was chosen, in a very transparent way. Please adapt this style to key point of the manuscript (e.g why Italy, or a certain method...). A small table could provide a better overview of the respective characteristics of each year (7/19-24) instead of bullet-points.

8/11: Please check grammar e.g. "All statistical analyses were performed…" Reference is needed to the ANOVA (Tamhane method).

8/15: "collected form all soil horizons". How many were there? Where is the supporting data and detailed portrayal of the soils?

8/16: Which are the main horizons? in 4.2 page 12 line 18 it is written A and B. What makes those two the main horizons?

Page 8/- Methods: In general are only few citation given in comparison of the many treatments/methods used e.g. Walkley & Black method, Tamhane method, Schwertmann method, Blakemore method. Please add the missing references.

8/7: When is a class defined as predominant and not exclusive? What is the threshold value? The same goes for continuous and discontinuous natural grassland. When to call which which?

8/15: how were those bulk soil samples collected? shuffle, soil corer, same 200cm3 cylinder?

8/18: What was the air temperature? How long were the samples dried them? Please add additional information and add precision to the method section and writing e.g. "The 2 mm fraction was used for further analyses." There should be no room for interpretation of how it was care bout and what was carried out.

8/19: please add literature reference to the "Walkley & Black method". Why was not a modern C/H/N measurement carried out?

8/21: Please add literature reference to the "Schwertmann method". Please add instrument specifications of the ICP-AES instrument. Please provide the full results of the measurements with the according standard error.

8/23: Very good that the Alox + ½ Feox, and the P retention were obtained. Please check again the WRB, are those really andic features? Please use the according WRB soil taxonomy. Provide a detailed soil description, at least of the horizon of interest. This includes (eventual the Munsell colours and ) clear differential of andic and vitric.

8/24: please add literature reference to oxalate method (I think Mizota and van Reeuwijk, (1989))  and the "Blakemore method".

9/ 7: Please add an article "the" to the word "use" or change to "using".

9/15 : the stated formula should have also been created with a formula editor, having the number (2).

10/9: Very good for providing a better consistency with the WRB classification compared to version 1 (9/9 year 2006, 8/7 year 2014). Please ensure that truly the right classification is used for all interpretations. Just changing the year numbering is not sufficient. Furthermore, please add the according year, to all WRB references in text, tables and figures.

10/12: What does a high and low Alo+0.5Feo% exactly say about the ordering? Figure 2 has it graphically explained, but a written explanation would be advisable. Use a similar writing style as in 7/13. Please, also add references in regard to the relation of this value and clay mineral ordering.

10 / 8-13.  Suggesting to split the sentence into 2. Second sentence could start in line 10 with  "Most interestingly…"

10/14: Above it is stated that 28+7 soils were investigated. How are those referred 42 pedons important? How do those mention 42 pedons connect with the 35 investigated once? Please add additional geographic information and explain "these pedons". Could be added to the sampling map. "horizon-based means" of what kind?

10/18: "dataset shows". There is not any full data set shows. There is only an essence of diverse datasets present in table 1.

10 / 22: What are the other "land uses"? Please specify.

11/1-4: Please re-arrange commas or restructure sentence.

11/7: Minor remark, delete space between "that" and the "comma".

11/8: … wetter year 2014? Please remind the reader what conditions defer 2003-2005 and 2014.

11/16-17. I suggest shortening the sentence into 3. e.g. ...the main features of the studied soils are reported. The soil… …  Moreover…

11/18: be careful with andic features. For the result section it is stated the WRB of 2006 in version 1. Now it is WRB of 2015. The diagnostic properties for andic soils of WRB 2006 would be Alox + 1/2 Feox of >2%, <0.9kg dm-3, P retention >85% and organic C of < 25%. Furthermore it would be sub-classified to sil-andic and alu-andic for certain values. In comparison to vitric, which would have an Alox + 1/2 Feox of >0,4% and a P retention of >25% etc. Please re-check the data, if it still fulfills the according requirement of the used year 2015, and add accuracy to this section.

11/18-19: Please provide the exact the P retention percentages. In table 2 retention values are provided, how does the reader know which % expresses which range, moderate or high? What does „high" mean? 85%, 90%, 66%.?

11/24: Maybe the findings with the used cover classes (Beech, Oak, Chestnut, Grassland) have shown, that there is no correlation / or non, could be found, but this is not in general the case. Recent finding e.g. the sampling strategy by Raab et al. (2017) clearly reads "ferns were used as bioindicators, as they are primarily found on acidic substrates, which is a common feature for volcanic soils." Therefore, if a vegetation cover is investigated, it should be considered that certain plants actual prefer an environment created by Andosols anyway. Are the used cover classes actual in favor of acidic soils? Would the vegetation even have the pre-requirement to even be considered to be used as correlating factor?

It could also simply just be re-phrase to "the investigated vegetation covers seem to be of little importance in determining…". Be careful with the wording "andic" as it requires clear parameters. Parameters, which cannot be seen, as there is no table that provides the according data.

12/8: Citation: Dumig or Dümig?

12/6-10. This is introduction information, which would fit very well at 3/22.

12/18: What are the selection criterias?

13/-: Please explain why only 16 data points were chosen for Fig.3., and how they were selected. Are those just A horizons of Table 3. If so, why?

The discussion reads more fluently compared to version 1. A refining of the key points that have to be emphasized would be welcomed. As well as a increase of precision and clarification of certain steps (e.g. why only 16 data points, high/low P retention etc). A grammar correction is still needed, as there are several sentences with wrong sentence structure (e.g. 14/15), missing articles (e.g. 14/16) etc. Please reinforce and emphasizes better what is different and what is similar from you're your work to the work of others. There still long paragraphs without references.

14/15: This key point is very well emphasized! Also previous arguing (/14/6-12) is very concise and clear!

14/9-10: The affect of land management is mentioned. Please provide in the tables what is/was the land use of the investigated soils and study areas.

15/6-11: A splitting of the sentence in two is recommended.

15/16-17: Please see comment 17/4. Looking at Table 2, it is seen that the Phaeozems have the highest value of organic C as also written on page 11 line 20-21. This should not be neglected. Furhther named soils which are do not show any values in any tables of this study should have at least an referenced value provided e.g. Regosols (___), Podzols (____)

16/4-5: "known to be easily erodible", by whom? Please add reference. Check the WRB criteria for "andic".

16/22: Which WRB?, Which Soil Taxonomy? Please add references.

17/4: C-storage and C-residence time are often mentioned (abstract, introduction, discussion, conclusions). Could accurate numbers be provided about it? Suggesting to provide the audience absolut numbers in the introduction ,page 3 line 7,from Post, 1983; 7 Batjes, 1996; Amundson, 2001.
* * *
References

Use same formatting for all references. Either shorten all forenames or non. e.g. Chen Yao or Dixon, R.K., —> follow SE guidelines.
Terribile, 2006 is missing in the reference list.
Cecchini, 2002 is in the reference list, but not used in the manuscript.
APAT is confusing as "CORINE land cover" is used in the text.

To ensure a bright diversity of scientific content it is important that a good mix of different sources is present in the paper. Looking through the reference list the following has been found:

Total reference#: 59
Corr. of missing: -2 +1
New number: 58

**of times named as Author / Co-author / Total / %**
Terribile: 2 / 5 / 7 / 12%
Iamarino: 2 / 1 /  3 / 5 %
Langella: 0 / 0 / 0 / 0%
Manna: 0 / 0 / 0 / 0%
Mileti: 0 / 1 / 1 /  2%
Vingiani: 2 / 1 / 3 / 5%
Basile: 3 / 2 / 5 / 8%
Cross overlapping sum: 9 / 10 / 19 / 33%

Without any overlaps as (co-)author, 8+1 paper of the authors are citied, equals in an amount of about 16% of self-citation.  I just want to provide these numbers for transparency reasons. As there are still numerous references of the method section still missing (as suggested to be added) I am very confident that this 16% will decrease drastically.
* * *
Figure captions:

Fig.1. What kind of map is that and what is the source of it? Please provide references.

Fig.2. Which WRB classification has been used? Mentioned in the text, but not at the figure caption.

Fig.3. Explain the differences of data points and sampling numbers in the text. Using just a selective number of data points, without reasoning in the text does not represent a scientific approach.

Fig.4. The weight mean Alo and Fe according to horizon thickness is used. It would be useful show also the horizon thickness of each pedon in a table, to make the results reproducible to other.
* * *
Figures

Fig.1. The map is definitely an improvement to version 1, where it was missing. Nevertheless, the color, and size of the marking points has to be reconsidered, as those are poorly readable. Also a geodetic system (e.g. WGS84), elevation scale, and a metric scale would improve the figure.

On page 6, line 17-20 it is written that were investigated 28 +7 soils= 35. On the map I have found only 28 triangles. Missing data points have to be added for transparency, as well as a legend, that makes it possible to differentiate the numerous sources. Further it is advised that the used classification of Fig.2: Umbrisol, Phaeozem, Cambisole and Andosol is added to see the distribution, using different markings and colors.

I am surprised that the Sila massif in Calabria has not been used in the study. Especially as F. Scarciglia (2008) is cited in the introduction. Scarciglia et al. (2008) wrote clearly about the volcanic soil formation in Calabria as well as the co-author Vingiani (2014). Please elaborate why the Sila massif was excluded in the study, as it clearly fulfills the NVME with an average elevation of 1300 m as (previous criteria >700 m) and low slope gradients?

Fig.2. What does each bar represent? I counted 36 bars, compared to the stated 35 samples (28+7, page 6 , line 17-20), therefore I assume each bar represents one investigation site. But, where does the Umbrisol bar come from? There is no reference made in the figure captions, nor in the graphic. Furthermore, as a reader it is not transparent which bar represent which sampling site market on the map (figure 1). Therefore it is not possible to identify which location shows which features and characteristics. A geographical reference to the presented data would improve the informational value for the reader.

Fig.3. Please explain why only 16 values are plotted. Based on Table 3 I assume that only the values of the A horizons are plotted. If so, please add this information and explain also why. Why were the B horizon values not plotted? Only 13 Andosols are presented in figure 2. Also in the manuscript text it is not clarified why only 16 values where chosen of a dataset with a sample size of 35.

Fig.4. Good overview of the data, although hard to read. An increase of the font size and symbol size is suggested, as there is still enough space in-between the individual graphics. Please explain also why there are only 8 Grassland land-covers shown in Table 1 and Table 2, but 9 points plotted for Grassland within all the selected years (2003,2005,2014).
* * *
Tables

Table1: Lack of detail information which of the soils comes from which data source. In section "2.2 Soil sampling" (page 6) several different sources are explained. Why is the data not displayed in groups containing all data and their according sources to have a full transparency? The average values can than be displayed on the bottom of each group. If a more comprehensive table is provided, it should also include the full coordinates of each investigate soil.

Table 2: Same poverty of displaying used data as in Table 1. Please create a more comprehensive, and even more importantly, transparent table. The shown table makes it difficult to reproduce the steps, nor are newly measured soil values accessible to the community. Why is only the A horizon mode displayed, although in section "4.2 Andic features and soil hydrology" (page 12) it is stated that "… a selection of .. .form horizon A and B of…"?

Please label the percentage sign (above the values 37, 69…) so readers know what the numbers actual represent. Please add to the supporting information of the table how the organic C, Alo+05Feo and P retention was evaluated (method name). What was the depth of the horizon? This is key information, as weighted values were created based on the horizon thickness.

Table 3. Here the horizons A and B are presented. It is absolutely unclear why a certain number of soils are selected for each horizon, nor are the individual values visible. Please clarify and provide a better transparency.

Table 4: Small graphic error of the line formatting (dotted line instead of straight line=), below first years row. Please add a reason why certain values are bold and others are not, so they audience understands their importance.
* * *
I hope the stated suggestions are useful, helpful and improve the quality of this manuscript. I did my very best to act as objectively as possible for the benefit of science.

---

## Short Comment (SC1) · 8 Sep 2017

Dear Referee#1, thank you for the helpful revision and the detailed comments you reported on the paper. We are now working in order to complete the reported dataset, for a better traceability of results, as well as to report the missing literature.

Best regards.

---

## Referee Comment (RC2) · Anonymous Referee #2 · 11 Sep 2017

Dear Editor and Authors,

I came through the manuscript 'The hidden ecological resource of andic soils in mountain ecosystems: evidences from Italy', which in my opinion represents a good and substantially novel evaluation of the role of soils with andic properties in non-volcanic mountain environments. Although the Authors evaluated only Italian soils, I agree with them that a similar approach could be successfully tested on other non-volcanic mountain environments worldwide, possibly after some methodological adaptations.

The manuscript is on the whole well-structured and well-written, but a number of minor typing corrections of the English language (e.g. single/plural, verb accordance, articles, etc.) are required. Some sentences are too long and articulated -Italian style- and difficult to read and hence should be shortened or simplified. Some other minor points

have to be clarified in the text. I marked many of them and gave some suggestions in my annotated pdf file, uploaded in the electronic platform.

In my opinion the overall scientific quality of the ms is good. However, I have a major point of concern (see also my annoteted pdf), regarding the fact that even though all the sampling sites were indeed in Italian non-volcanic mountain environments, most of them are close to volcanoes, which have been active during Plio-Quaternry times (e.g. Peccerillo, 2005). This point needs to be clearly stated in the rationale behind the objectives of the ms and has to be commented in the Discussion.

As regarding the Conclusions section, it should be more concise and to the point. Moreover, it includes conclusions along with open questions. I suggest to separate these issues in two subsections or change the heading of the section (e.g. Conclusive remarks and future perspectives, or Conclusions and open questions, or something similar).

Figures and tables are of good quality, but geographic coordinates (Lat and Long) and the place names reported in the text (Alps, Apennines, Monte Bianco, etc.) have to be included in the map of figure 1.

Based on the above considerations I'd recommend a minor revision prior to resubm the manuscript.

Best regards,

Please also note the supplement to this comment:
https://www.solid-earth-discuss.net/se-2017-57/se-2017-57-RC2-supplement.pdf

―――――――――――――――――

**Supplement:**

[revised manuscript text omitted]

---

## Author Comment (AC1) · 1 Nov 2017

Dear reviewer, we addressed systematically all your requests. In the attached pdf files (as supplement) you will find the answer for each comment. We report the reviewer comments followed by our answers (in bold). As figure files you will find i) marked version of the manuscript (comprehensive of the corrections asked by the referee 2) ii) amended version of the manuscript. Thank you for your detailed work of revision that implemented the quality of the paper.

Kind regards Michela Iamarino

Please also note the supplement to this comment:

[Figure]

https://www.solid-earth-discuss.net/se-2017-57/se-2017-57-AC1-supplement.pdf

1  **The hidden ecological resource of andic soils in mountain ecosystems:**

2  **evidences from Italy**

3  Fabio Terribile[1,2], Michela Iamarino[1*], Giuliano Langella[3], Piero Manna[3],

4  Florindo Antonio Mileti[1], Simona Vingiani[1,2], Angelo Basile[2,3]

5  [1]*Department of Agricultural Sciences, University of Naples Federico II, Via Università*

6  *100, 80055 Portici (Napoli), Italy*

7  [2] *Interdepartmental Research Centre CRISP, University of Naples Federico II, Via*

8  *Università 100, 80055 Portici (Napoli), Italy*

9  [3] *CNR ISAFOM, Via Patacca 85, 80056 Ercolano (Napoli), Italy*

10  *corresponding author: terribilesci@gmail.com

11

12  **Abstract**

13  Andic soils have unique morphological, physical and chemical properties that induce

14  both considerable soil fertility and great vulnerability to land degradation. Moreover

15  they are the most striking mineral soils in terms of large organic C storage and long C

16  residence time. This is especially related to the presence of poorly crystalline clay

17  minerals and metal-humus complexes. Recognition of andic soils is then very important.

18  Here we attempt to show, through  a combined analysis of 35 sampling points

19  chosen  in accordance to

20  specific physical and vegetation rules, that some andic soils

21   have an utmost ecological importance.

22  More specifically, in Italian  non-volcanic mountain ecosystems (>  600 m)

23  combining  low slope  locations (< 21%) and highly active green

**Fig. 1.** Author's changes in the manuscript

1  **The hidden ecological resource of andic soils in mountain ecosystems:**

2  **evidences from Italy**

3  Fabio Terribile[1,2], Michela Iamarino[1*], Giuliano Langella[3], Piero Manna[3],

4  Florindo Antonio Mileti[1], Simona Vingiani[1,2], Angelo Basile[2,3]

5  [1]*Department of Agricultural Sciences, University of Naples Federico II, Via Università*

6  *100, 80055 Portici (Napoli), Italy*

7  [2] *Interdepartmental Research Centre CRISP, University of Naples Federico II, Via*

8  *Università 100, 80055 Portici (Napoli), Italy*

9  [3] *CNR ISAFOM, Via Patacca 85, 80056 Ercolano (Napoli), Italy*

10  *corresponding author: terribilesci@gmail.com

11

12  **Abstract**

13  Andic soils have unique morphological, physical and chemical properties that induce

14  both considerable soil fertility and great vulnerability to land degradation. Moreover

15  they are the most striking mineral soils in terms of large organic C storage and long C

16  residence time. This is especially related to the presence of poorly crystalline clay

17  minerals and metal-humus complexes. Recognition of andic soils is then very important.

18  Here we attempt to show, through the a combined analysis of 35 sampling points

19  chosen, throughout the Italian non volcanic mountain landscapes, in accordance to

20  specific physical and vegetation rules, that some andic soils rich in poorly crystalline

21  clay minerals have an utmost ecological importance.

22  More specifically, in Italian various non-volcanic mountain ecosystems (> 700 600 m)

23  combining and in low slope gradient locations (< 21%12°) and highly active green

**Fig. 2.** Amended version of the manuscript

**Supplement:**

**REFEREE#1**

1 / 17: Suggestion, dot after "time". New sentence starting: This feature/unique capability is especially…

**Done**

1 / 18: Specify "these". e.g. delete "these" and write "andic"

**Done**

1 / 19-22: sentence with 41 words should be cut in half, to improve the readability.

**Done**

2/ 2: What is really define as low slope in this study? 12° or 30°, as later on page 6 line 12 written? Please clarify the used parameters

**Done (21%)**

2 / 2: Please do a thorough grammar correction of the manuscript for quality reasons e.g. "an unique" not "a unique"

**Done**

3 / 3: As state before, please do a sincere grammar correction e.g. delete "soil" in front of morphological and put it before properties

**Done**

3/3 : Please be precise with the wording e.g. instead of "Between" use "Among or The main characteristics are.."

**Done**

3 / 6-12: To improve readability, please split this sentence.

**Done**

3 / 17: a single "artefact" or multiple "artefacts" ?. Also a new sentence could be started here. As a sentence with >50 words seem a bit long (line 12-18) for most readers.

**Done**

3/ 22: it would be helpful for readers to write the countries and regions where the according authors (as done at 12/6-10) have found volcanic landscapes in non-volcanic mountain ecosystems, as „throughout the world" is very general. This would also emphasis the impact/importance of this research. Please state why Italy was chosen as study object, and not any of those other "around the world"?

**Done**

4/1-3: Please provide clearer information about the C storage capacity. It is stated that andic soils

have important C storage abilities. Which ones? How much more or longer can they store it? Why kind of soils so special?

**Done**

4/20: Please clarify parameter for "low slope", 12° or 30° as later on page 6 line 12 written?

**Done**

5/3: instead of ":" a dot would be sufficient

**Done**

5/13: In the introduction it is explained that NVME are found around the world. Why is the focus than on Italy? Maybe start with the necessary features and narrow it than down to the best fitting location, which seems to be Italy in this case. Please provide references of the provided information within the section "study site".

**Done**

5/13: Very well that there is now a map added, compared to version 1.

**OK**

6/8: how was the sampling carried out. What material was used to do the sampling? An additional table including all sampling sites, with their slopes, soil type, horizon depth, elevation etc. would improve the transparency that each chosen location fulfilled the designed requirements. This is not the case yet, as only average values are presented.

**Soil profiles were dug, described and sampled following FAO 2006. It has been specified in the paragraph "Methods". The table S1 (supplementary material) was added to the manuscript. We reported slopes, soil classification, soil depth, elevation, aspect, land use and location for each sampling site.**

6/10: Please clarify parameters. >600m or >700m as stated previously (page 2, line 1)

**Done. The vast majority of sites are above 600 m. In addition we inserted 5 points in the landscape (<21%, max NDVI >0,65) below 600 m. This was useful to have a larger dynamic range (e.g. r analysis) of elevation and andic features**

6/12: As mentioned early, please clarify e.g. 30° or 12°

**Done (21%)**

6/18: please explain briefly the methodology. As the work in this study is with 28 samples mainly based on data of the paper "Iamarino and Terribile, 2008". It is suggested that key data, which were used for this evaluation should be presented in a referenced table.

**36 is the total number. 28 are data from Iamarino and Terribile (2008) and Iamarino (2005); 6 are newly surveyed soils, 1 from Frezzotti and Narcisi (1996), and 1 from ISRIC (2005). In table S2 (supplementary material) we report horizon sequence, color of A and B horizons, weighted mean among the soil horizons for organic carbon, Alo+0.5Feo and P retention, max value of**

**Alo+0.5Feo.**

6/21: Please do a grammar check, preposition is missing after "...and 5 soils…"
**Done**

page 5-6: The clearer separation of the soil sampling and the computational part is an improvement to version 1.

**OK**

7/13-24: A good, logic and clear-presented paragraph, that emphasis in detail „why", „what" was chosen, in a very transparent way. Please adapt this style to key point of the manuscript (e.g why Italy, or a certain method...). A small table could provide a better overview of the respective characteristics of each year (7/19-24) instead of bullet-points.

**We have rephrased the paragraph making it more logic and smooth. We would rather prefer to avoid to include an extra table considering that we show and comment only very few data.**

8/11: Please check grammar e.g. "All statistical analyses were performed…" Reference is needed to the ANOVA (Tamhane method).

**Done**

8/15: "collected from all soil horizons". How many were there? Where is the supporting data and detailed portrayal of the soils?

**126 soil horizons (we added this information in the text). Data are reported in Table S2.**

8/16: Which are the main horizons? in 4.2 page 12 line 18 it is written A and B. What makes those two the main horizons?

**As shown in the column "horizon sequence" in table S2, the main horizons are topsoil A horizons and subsoil Bw horizons.**

Page 8/- Methods: In general are only few citation given in comparison of the many treatments/methods used e.g. Walkley & Black method, Tamhane method, Schwertmann method, Blakemore method. Please add the missing references.

**Added**

8/7: When is a class defined as predominant and not exclusive? What is the threshold value? The same goes for continuous and discontinuous natural grassland. When to call which which?
**Done**

8/15: how were those bulk soil samples collected? shuffle, soil corer, same 200cm3 cylinder?
**Bulk samples were collected by a trowel. Steel cylinders of about 200 $cm^3$ were carefully inserted in the selected A and Bw horizons by an impact absorbing hammer. Accordingly, we changed the text in the manuscript.**

8/18: What was the air temperature? How long were the samples dried them? Please add additional

information and add precision to the method section and writing e.g. "The 2 mm fraction was used for further analyses." There should be no room for interpretation of how it was care bout and what was carried out.

**Air temperature and time of drying was added. I am not sure about the interpretation of your statement "There should be no room for interpretation of how it was care bout and what was carried out", however for the sake of clarity, in our analyses we made duplicate chemical analyses for each sampled soil as quality control routine in our lab. Then our results are an average of two duplicate measures.**

8/19: please add literature reference to the "Walkley & Black method". Why was not a modern C/H/N measurement carried out?

**Added literature. The original procedure (Walkley-Black procedure) was Walkley and Black, 1934; nevertheless the method actually used in the chemical lab refers to a revision made by Walkley in the 1947. Walkley, A. A critical examination of a rapid method for determining organic carbon in soils - effect of variations in digestion conditions and of inorganic soil constituents. *Soil Sci.* 63: 251-265, 1947.**

**This is the method followed by WRB for soil classification. However, other procedures, including carbon analysers (e.g. dry combustion) may also be used.**

8/21: Please add literature reference to the "Schwertmann method". Please add instrument specifications of the ICP-AES instrument. Please provide the full results of the measurements with the according standard error.

**Reference for the Schwertmann and Blakemore methods were added, as well as ICP-AES specification.**

**About the measurements and the according standard errors, in our analyses we made duplicate chemical analyses for each sampled soil as quality control routine in our lab. Then our results are an average of two duplicate measures.**

8/23: Very good that the Alox + ½ Feox, and the P retention were obtained. Please check again the WRB, are those really andic features? Please use the according WRB soil taxonomy. Provide a detailed soil description, at least of the horizon of interest. This includes (eventual the Munsell colours and ) clear differential of andic and vitric.

**We have reformulated it and moreover we inserted the new table in supplementary materials with required data. We now provide these data in Table S2. We also rechecked the WRB soil classification**

8/24: please add literature reference to oxalate method (I think Mizota and van Reeuwijk, (1989)) and the "Blakemore method".

**We added both Schwertmann, 1964 and Blakemore et al., 1987**

9/ 7: Please add an article "the" to the word "use" or change to "using".
**Done**

9/15 : the stated formula should have also been created with a formula editor, having the number (2).

**Done**

10/9: Very good for providing a better consistency with the WRB classification compared to version 1 (9/9 year 2006, 8/7 year 2014). Please ensure that truly the right classification is used for all interpretations. Just changing the year numbering is not sufficient. Furthermore, please add the according year, to all WRB references in text, tables and figures.

**Done**

10/12: What does a high and low Alo+0.5Feo% exactly say about the ordering? Figure 2 has it graphically explained, but a written explanation would be advisable. Use a similar writing style as in 7/13. Please, also add references in regard to the relation of this value and clay mineral ordering.

**According with this remark, we reported ranges of Alo+0.5Feo% and references.**

10 / 8-13. Suggesting to split the sentence into 2. Second sentence could start in line 10 with "Most interestingly…"

**Done**

10/14: Above it is stated that 28+7 soils were investigated. How are those referred 42 pedons important? How do those mention 42 pedons connect with the 35 investigated once? Please add additional geographic information and explain "these pedons". Could be added to the sampling map. "horizon-based means" of what kind?

**We deleted the 42 issues because it may generate in the reader some confusion**

10/18: "dataset shows". There is not any full data set shows. There is only an essence of diverse datasets present in table 1.

**Done**

10 / 22: What are the other "land uses"? Please specify.

**Done**

11/1-4: Please re-arrange commas or restructure sentence.

**Done**

11/7: Minor remark, delete space between "that" and the "comma".

**Done**

11/8: … wetter year 2014? Please remind the reader what conditions defer 2003-2005 and 2014.

**Done**

11/16-17. I suggest shortening the sentence into 3. e.g. ...the main features of the studied soils are reported. The soil… … Moreover…

**Done**

11/18: be careful with andic features. For the result section it is stated the WRB of 2006 in version 1. Now it is WRB of 2015. The diagnostic properties for andic soils of WRB 2006 would be Alox + 1/2 Feox of >2%, <0.9kg dm-3, P retention >85% and organic C of < 25%. Furthermore it would be sub-classified to sil-andic and alu-andic for certain values. In comparison to vitric, which would have an Alox + 1/2 Feox of >0,4% and a P retention of >25% etc. Please re-check the data, if it still fulfills the according requirement of the used year 2015, and add accuracy to this section.

**Done, we rechecked data according with requirement of the IUSS Working Group WRB, 2105**

11/18-19: Please provide the exact the P retention percentages. In table 2 retention values are provided, how does the reader know which % expresses which range, moderate or high? What does „high" mean? 85%, 90%, 66%.

**Done**

11/24: Maybe the findings with the used cover classes (Beech, Oak, Chestnut, Grassland) have shown, that there is no correlation / or non, could be found, but this is not in general the case. Recent finding e.g. the sampling strategy by Raab et al. (2017) clearly reads "ferns were used as bioindicators, as they are primarily found on acidic substrates, which is a common feature for volcanic soils." Therefore, if a vegetation cover is investigated, it should be considered that certain plants actual prefer an environment created by Andosols anyway. Are the used cover classes actual in favor of acidic soils? Would the vegetation even have the pre-requirement to even be considered to be used as correlating factor?

**Done (we could not find Raab but we found a similar published work by Ciarkowska, K; Miechowka, A. 2017).**

It could also simply just be re-phrase to "the investigated vegetation covers seem to be of little importance in determining…". Be careful with the wording "andic" as it requires clear parameters. Parameters, which cannot be seen, as there is no table that provides the according data.

**In the tables S1 and S2 we reported data that enable to check the andic properties**

12/8: Citation: Dumig or Dümig?

**Done. Dümig**

12/6-10. This is introduction information, which would fit very well at 3/22.

**Done**

12/18: What are the selection criterias? In the text "In order to address this issue, a selection of undisturbed soil samples, from horizons A and B, of the previously investigated soils were analysed."
13/-: Please explain why only 16 data points were chosen for Fig.3., and how they were selected. Are those just A horizons of Table 3. If so, why?

**As it is wellknown, soil hydrology measurements (such as the water retention curve measured in**

**our samples) are very time-consuming and expensive. Then it is always a must a careful soil samples selection. Then in this study we sampled only selected a and bw horizons in representative pedons of the soil types (Andosols, Cambisols and Phaeozem) encountered in our investigation given in TABLE S2.**

The discussion reads more fluently compared to version 1. A refining of the key points that have to be emphasized would be welcomed. As well as an increase of precision and clarification of certain steps (e.g. why only 16 data points, high/low P retention etc). A grammar correction is still needed, as there are several sentences with wrong sentence structure (e.g. 14/15), missing articles (e.g. 14/16) etc. Please reinforce and emphasizes better what is different and what is similar from you're your work to the work of others. There still long paragraphs without references.

**The lack of references is due to the evidence that there is a lack of scientific literature concerning the relationship between andic soil and remotely sensed vegetation indices.**

14/15: This key point is very well emphasized! Also previous arguing (/14/6-12) is very concise and clear!
**OK**

14/9-10: The effect of land management is mentioned. Please provide in the tables what is/was the land use of the investigated soils and study areas.

**We provide now the land use in table S1 for each study area.**

15/6-11: A splitting of the sentence in two is recommended.
**Done**

15/16-17: Please see comment 17/4. Looking at Table 2, it is seen that the Phaeozems have the highest value of organic C as also written on page 11 line 20-21. This should not be neglected. Furhther named soils which are do not show any values in any tables of this study should have at least an referenced value provided e.g. Regosols (___), Podzols (____)

**Done**

16/4-5: "known to be easily erodible", by whom? Please add reference. Check the WRB criteria for "andic".

**The sentence has been changed in "andic soil are known to be among the most vulnerable soils in the world in terms of soil erosion". Full details were provided in the introduction section**

16/22: Which WRB?, Which Soil Taxonomy? Please add references.
**Done**

17/4: C-storage and C-residence time are often mentioned (abstract, introduction, discussion, conclusions). Could accurate numbers be provided about it? Suggesting to provide the audience absolut numbers in the introduction ,page 3 line 7,from Post, 1983; 7 Batjes, 1996; Amundson, 2001.

**We reported data in the introduction section**

References

Use same formatting for all references. Either shorten all forenames or non. e.g. Chen Yao or Dixon, R.K., —> follow SE guidelines.

**Done**

Terribile, 2006 is missing in the reference list.

**We did not find in the text**

Cecchini, 2002 is in the reference list, but not used in the manuscript.

**Deleted**

APAT is confusing as "CORINE land cover" is used in the text.

**Amended**

Figure captions:

Fig.1. What kind of map is that and what is the source of it? Please provide references.

**The source is a dtm, but we used it as a sketch map as usually done in many other published papers to locate sites**

Fig.2. Which WRB classification has been used? Mentioned in the text, but not at the figure caption.

**IUSS Working Group WRB, 2015. Added in the captions**

Fig.3. Explain the differences of data points and sampling numbers in the text. Using just a selective number of data points, without reasoning in the text does not represent a scientific approach.

**This issue has been explained in the text**

Fig.4. The weight mean Alo and Fe according to horizon thickness is used. It would be useful show also the horizon thickness of each pedon in a table, to make the results reproducible to other.
In table s2 is provided the soil depth of each pedon. We think this is enough to understand the meaning of weighted mean.

**This paper deals with the relationship between NDVI metrics and andosolization processes as estimated by Alo+0.5Feo. It is not a paer discussing andic soil database in Italy. To this respect, to address the very detailed reviewer requests, we indeed included in S1 and S2 tables many analytical data on investigated pedons, but we are not providing the entire database (depth of each horizon, color, soil structure, texture, etc.)**
* * *
Figures

Fig.1. The map is definitely an improvement to version 1, where it was missing. Nevertheless, the color, and size of the marking points has to be reconsidered, as those are poorly readable. Also a geodetic system (e.g. WGS84), elevation scale, and a metric scale would improve the figure.

**We used a standard approach (Dem based sketch of Italy) to locate the sampling points. We think that further additional information to be provided in this figure will confuse the reader. Anyway all required information are provided in the tables.**

On page 6, line 17-20 it is written that were investigated 28 +7 soils= 35. On the map I have found only 28 triangles. Missing data points have to be added for transparency, as well as a legend, that makes it possible to differentiate the numerous sources. Further it is advised that the used classification of Fig.2: Umbrisol, Phaeozem, Cambisol and Andosol is added to see the distribution, using different markings and colors.

**As it is shown in the table S1 of the additional material, some locations are very cose to each other. Then, they overlap when sketched in the figure 1. We have improved the figure caption to explain this specific point to the reader. This paper is focused on the ecological relevance of andic soils estimated BY Alo+0.5Feo. Then soil type is not the main parameter and its inclusion in the first may mislead the reader.**

I am surprised that the Sila massif in Calabria has not been used in the study. Especially as F. Scarciglia (2008) is cited in the introduction. Scarciglia et al. (2008) wrote clearly about the volcanic soil formation in Calabria as well as the co-author Vingiani (2014). Please elaborate why the Sila massif was excluded in the study, as it clearly fulfills the NVME with an average elevation of 1300 m as (previous criteria >700 m) and low slope gradients?

**In Sila massif there are wonderfull andosols, but unfortunately they occur under pine forest which is a type of land use not addressed by our specific paper. Nevertheless, we take on-board this advice for future studies.**

Fig.2. What does each bar represent? I counted 36 bars, compared to the stated 35 samples (28+7, page 6 , line 17-20), therefore I assume each bar represents one investigation site. But, where does the Umbrisol bar come from? There is no reference made in the figure captions, nor in the graphic. Furthermore, as a reader it is not transparent which bar represent which sampling site market on the map (figure 1). Therefore it is not possible to identify which location shows which features and characteristics. A geographical reference to the presented data would improve the informational value for the reader.

**Done in table S2, in this figure we seek to provide an immediate visual assessment of Al+0.5Feo variation against soil type. This is the why we rank in increasing order Al+0.5Feo against soil type. We deleted the umbrisol soil type (it was a mistake). Requested data are provided in the supplementary materials**

Fig.3. Please explain why only 16 values are plotted. Based on Table 3 I assume that only the values of the A horizons are plotted. If so, please add this information and explain also why.

Why were the B horizon values not plotted? Only 13 Andosols are presented in figure 2. Also in the manuscript text it is not clarified why only 16 values where chosen of a dataset with a sample size of 35.

**In order to avoid to compare different physical system, we plotted only A horizons (horizon reach in organic matter) and not Bw horizons (horizon poor in organic matter). The plot of the only A horizons refer also to their ecological importance, since they are the topsoil, then the soil horizon mostly affecting the soil fertility.**

**Figure 3 shows the relationship between IRI and Alo+0.5Feo (in the range 0.4-2.5%). This has nothing to do with soil types (e.g. 13 Andosols quoted by the reviewer) but rather with andosolization process estimated by Alo+0.5Feo.**

Fig.4. Good overview of the data, although hard to read. An increase of the font size and symbol size is suggested, as there is still enough space in-between the individual graphics. Please explain also why there are only 8 Grassland land-covers shown in Table 1 and Table 2, but 9 points plotted for Grassland within all the selected years (2003,2005,2014).

**Corrections done**
* * *
Tables

Table1: Lack of detail information which of the soils comes from which data source. In section "2.2 Soil sampling" (page 6) several different sources are explained. Why is the data not displayed in groups containing all data and their according sources to have a full transparency? The average values can then be displayed on the bottom of each group. If a more comprehensive table is provided, it should also include the full coordinates of each investigate soil.

**We provided supplementary materials, tables S1 and S2, with additional soil data.**

Table 2: Same poverty of displaying used data as in Table 1. Please create a more comprehensive, and even more importantly, transparent table.

The shown table makes it difficult to reproduce the steps, nor are newly measured soil values accessible to the community.

**We provided supplementary materials, tables S1 and S2, with additional soil data.**

Why is only the A horizon mode displayed, although in section "4.2 Andic features and soil hydrology" (page 12) it is stated that "… a selection of .. .form horizon A and B of…"?

**Actually the plotting required the entire water retention curve that we had only for a subset (19 horizons) of the main representative A and B horizons.**

Please label the percentage sign (above the values 37, 69…) so readers know what the numbers actual represent. Please add to the supporting information of the table how the organic

C, Alo+05Feo and P retention was evaluated (method name). What was the depth of the horizon? This is key information, as weighted values were created based on the horizon thickness.

**Done**

Table 3. Here the horizons A and B are presented. It is absolutely unclear why a certain number of soils are selected for each horizon, nor are the individual values visible. Please clarify and provide a better transparency.

**In the new table S1 the selection of pedon and in the text is provided the reasoning of the text**

Table 4: Small graphic error of the line formatting (dotted line instead of straight line=), below first years row. Please add a reason why certain values are bold and others are not, so they audience understands their importance.

**Done. Bold character was deleted.**

---

## Author Comment (AC2) · 1 Nov 2017

Dear reviewer, we addressed systematically all your requests. In the attached pdf files (as supplement) you will find the answer for each comment. We report the reviewer comments followed by our answers (in bold). As figure files you will find i) amended version of the manuscript. ii) marked version of the manuscript (comprehensive of the corrections asked by the referee 1)

Thank you for your detailed work of revision that implemented the quality of the paper.

Kind regards Michela Iamarino

Please also note the supplement to this comment:

https://www.solid-earth-discuss.net/se-2017-57/se-2017-57-AC2-supplement.pdf

**The hidden ecological resource of andic soils in mountain ecosystems:**

**evidence from Italy**

Fabio Terribile[1,2], Michela Iamarino[1*], Giuliano Langella[3], Piero Manna[3],

Florindo Antonio Mileti[1], Simona Vingiani[1,2], Angelo Basile[2,3]

[1]*Department of Agricultural Sciences, University of Naples Federico II, Via Università*

*100, 80055 Portici (Napoli), Italy*

[2] *Interdepartmental Research Centre CRISP, University of Naples Federico II, Via*

*Università 100, 80055 Portici (Napoli), Italy*

[3] *CNR ISAFOM, Via Patacca 85, 80056 Ercolano (Napoli), Italy*

*corresponding author: terribilesci@gmail.com

**Abstract**

Andic soils have unique morphological, physical and chemical properties that induce both considerable soil fertility and great vulnerability to land degradation. Moreover they are the most striking mineral soils in terms of large organic C storage and long C

residence time. This is especially related to the presence of poorly crystalline clay minerals and metal-humus complexes. Recognition of andic soils is then very important.

Here we attempt to show, through a combined analysis of 35 sampling points chosen in accordance to specific physical and vegetation rules, that some andic soils have an utmost ecological importance.

More specifically, in Italian non-volcanic mountain ecosystems (> 600 m) combining low slope (< 21%) and highly active green biomass (high NDVI values) and in agreement to recent findings, we found the widespread occurrence of andic soils having

**Fig. 1.** amended version of the manuscript

**The hidden ecological resource of andic soils in mountain ecosystems:**

**evidences from Italy**

Fabio Terribile[1,2], Michela Iamarino[1*], Giuliano Langella[3], Piero Manna[3],

Florindo Antonio Mileti[1], Simona Vingiani[1,2], Angelo Basile[2,3]

[1]*Department of Agricultural Sciences, University of Naples Federico II, Via Università*

*100, 80055 Portici (Napoli), Italy*

[2] *Interdepartmental Research Centre CRISP, University of Naples Federico II, Via*

*Università 100, 80055 Portici (Napoli), Italy*

[3] *CNR ISAFOM, Via Patacca 85, 80056 Ercolano (Napoli), Italy*

*corresponding author: terribilesci@gmail.com

**Abstract**

Andic soils have unique morphological, physical and chemical properties that induce both considerable soil fertility and great vulnerability to land degradation. Moreover they are the most striking mineral soils in terms of large organic C storage and long C

residence time. This is especially related to the presence of poorly crystalline clay minerals and metal-humus complexes. Recognition of andic soils is then very important.

Here we attempt to show, through  a combined analysis of 35 sampling points chosen , in accordance to specific physical and vegetation rules, that some andic soils have an utmost ecological importance.

More specifically, in Italian  non-volcanic mountain ecosystems (>  600 m)

combining  low slope  locations (< 21%) and highly active green

**Fig. 2.** author's changes in the manuscript

**Supplement:**

**REFEREE#2**

4/16 The terms 'evaluating... soils' is too vague and generic. Please, explain which features (field, chemical, physical, etc.) of these 35 soils you evaluated.

**Done**

5/13 Put the geographic coordinates (Lat and Long) and the place names reported in the text (Alps, Apennines, Monte Bianco, etc.) in the map of fig. 1, for readers who are not familiar with Italy.

**In Table S1 and S2 we now report location, elevation, slope, aspect for each of the 35 analysed soils.**

6/1 Instead of mild - warm temperate (Mediterranean-type?)

A reference to climate classification would be welcome here. I suggest the following:

Kottek, M., Grieser, J., Beck, C., Rudolf, B., Rubel, F., 2006. World Map of the Köppen-Geiger climate classification updated. Meteorologische Zeitschrift 15 (3), 259-263

**Done**

6/10 On page 2, line 1 and page 4, line 19, you put another altitude limit of > 700 m asl. Please homogenize this datum throughout the text.

**Done**

6/15 Add: based on maximum value compositing (MVC) techniques (do you mean this tool?)

**We put full reference to the well known MVC method**

6/19 Put "additional" after 7.

**Done**

7/4 VI MVC has to be integrated as: vegetation index (VI) based on maximum value compositing (MVC) techniques.

**Done**

7/14 delete dashes first and after "potentially".

**Done**

8/3 The Corine land cover (CLC level 4, 5) classification… after CLC there is hierarchically.

**Done**

12/22 "Explicit" ???

**Re-phrased**

15/16-19: "Indeed, deep andic soils (as reported in this study) have about twice (Batjes, 1996) the mean organic C content of deep Regosols, Cambisols and Podzols **which** previous soil inventories (Mancini, 1966; EuDASM, 2007) considered as the main soil types in the investigated landscapes".

**Re-phrased**

Change which with "of" o "from"

**Re-phrased**

15/21 the referee suggests to change "Conclusion" saying:

This section includes some real conclusions together with a number of open questions. I suggest to change the section title, e.g. as Conclusive remarks and future perspectives, or Conclusions and open questions, or something similar) or separate two subsections. In addition Conclusions should more concise and to the point.

**Done**

16/4 Add "and surrounding piedmont zones" after "protection in mountain environment".

**Inserting the piedmont may open-up large questions since piedmont can include very different areas with different shapes and slopes**

17/3 "this key mineral soil" …These terms are ambiguous, because you are not referring to a unique soil group, but rather to a set of different soils types. Please, rephrase (e.g.: this key mineral soil with andic properties..., or: these key andic soils...) .

**We prefer to add (i.e. andic soils)…**

17/ 4 Can you give some orders of magnitude (and suited references), here?

**We answered at the same question at referee#1**